# A Light Recipe including Far-Red Wavelength during Healing of Grafted Watermelon Seedlings Enhances the Floral Development and Yield Earliness

**Filippos Bantis** [1,*] **, Anna Gkotzamani** [1] **, Christodoulos Dangitsis** [2] **and Athanasios Koukounaras** [1]

[1] Department of Horticulture, Aristotle University, 54124 Thessaloniki, Greece; agkotzam@agro.auth.gr (A.G.); thankou@agro.auth.gr (A.K.)

[2] Agris S.A., Kleidi, 59300 Imathia, Greece; cdaggitsis@agris.gr

[*] Correspondence: fbantis@agro.auth.gr

**Abstract:** Watermelon is widely propagated through grafting, after which seedlings are subjected to healing under controlled conditions including artificial lighting. Light wavelengths, such as blue, red, and far-red, impose considerable effects on seedlings, which possibly carry on to the mature plants. The aim of the present study is to examine whether different light wavelengths during healing of grafted watermelon seedlings impose variable effects during field cultivation. After grafting, seedlings were healed in an environmentally controlled healing chamber under fluorescent (FL) lamps and light-emitting diodes, providing 100% red (R), 100% blue (B), 88/12% R/B (12B), and 12B including 5% far-red (12B + FR). After acclimatization, seedlings were transplanted in the field. Vegetative growth until floral initiation was enhanced by 12B and 12B + FR, as shown by stem diameter and leaf number measurements. Flowering was mainly accelerated by 12B + FR and considerably decelerated by FL and B. The same pattern was followed by fruit yield, which was similar for all treatments at the end of the experiment. Nevertheless, fruit quality was not affected by any of the light treatments. It is concluded that a light recipe, including red, blue and far-red, wavelengths during healing of grafted seedlings enhances the overall growth, and flowering and yield earliness of watermelon crops.

**Keywords:** *Citrullus lanatus*; nursery; healing chamber; transplantation; photomorphogenesis; flowering; crop production; antioxidants





## 1. Introduction

Watermelon (*Citrullus lanatus* L.) is one of the most cultivated species among the *Cucurbitaceae* family worldwide. It is also one of the most exported horticultural species, mainly due to early harvests in some regions of southwest Greece. On average, watermelon yield in Greece reached more than 45 ton/ha during 2016–2020 and its export value has risen to over 52 million euros. In 2020, the area harvested was 8770 ha reaching productivity of 49.1 t/ha. The importance of the crop is highlighted by the export profit in that period, which constituted 12% of the total watermelon exported in Europe (FAOSTAT, 2022). In 2020, the total export value in Greece was over 60 million euros.

Watermelon is susceptible to soilborne pathogens, and, thus, it is grafted mostly onto gourds (*Lagernaria siceraria* Standl.) or onto interspecific hybrids (*C. maxima* Duch. × *C. moschata* Duch.) [1]. Healing is the most delicate stage of grafted watermelon seedlings production. During healing, seedlings must be grown in an environmentally controlled space, ideally a growth (healing) chamber, where temperature, relative humidity, and light are fully adjusted.

Light is a key factor in plant growth and development. Light acts through its characteristics, such as photoperiod (duration of emission), quantity (intensity), and quality (wavelength). Photosynthetically active radiation (PAR, 400–700 nm) is the part of visible light which is utilized by the plants for photosynthesis. Blue (425–490 nm) and red light

(620–700 nm) in particular are crucial for plant growth and development, increased yield, and fruit quality [2]. Far-red light (700–780 nm) is also important in the plant's life as it affects biological functions besides photosynthesis, such as seed germination, phototropism, and flowering [3]. In a recent publication involving hydroponically grown lettuce, Zhen and Bugbee (2020) [4] reported that far-red photons are equally effective for photosynthesis when acting synergistically with PAR photons.

Furthermore, climate change is an inevitable process that is expected to alter the prevailing environmental conditions. According to the 6th IPCC report (2021) [5], heat waves and drought periods will increase in frequency, thus increasing the risk for open-field vegetable crops to suffer from heat stress while also increasing their need for water. Increased heat is expected to deteriorate the abiotic stress of vegetable crops leading to the introduction of additional agrochemicals to cope with pests and pathogens, while the yields will mostly decline, thus raising the cost of production. Moreover, the increased water requirements will also lead to a production cost raise. To this end, the application of cultivation methods that reduce the growing cycle (i.e., production time) even for a few days is crucial in economic terms. This includes the earlier coverage of market demand, leading to higher income for all participants in the supply chain, from growers to retailers.

There is complete lack of literature regarding the effect of different light-emitting diode (LED) light spectra on the flowering, yield, and quality of watermelon crops. Previous studies of our group [6,7] highlighted the importance of red and blue wavelengths for the production of vigorous, high-quality grafted watermelon seedlings, which points to why we used such wavelengths in our research. Moreover, cucumber transplants illuminated with various wavelengths exhibited an after-effect during flowering and harvest [8]. Our research hypothesis was that light quality influences vegetative growth during the first few weeks after transplantation, and possibly affects flowering since the first flower buds differentiate during the nursery growth. Therefore, our objective was to examine whether different light wavelengths during the healing of grafted watermelon seedlings impose variable effects during field cultivation. To this end, our efforts focused on evaluating the plant development, the flowering, the yield, and fruit quality with the aim of increasing the fruit earliness.

## 2. Materials and Methods

### 2.1. Plant Material and Grating

The grafted seedlings were produced in the facilities of Agris S.A. in Kleidi, Imathia, Greece. Watermelon hybrid scions (*Citrullus lanatus* L., Celine F1) and interspecific squash hybrid rootstocks (*Cucurbita maxima* × *C. moschata*, TZ-148) were cultivated according to standard commercial practices. For a detailed description of this stage of cultivation, please refer to Bantis et al. [6]. When the scion and rootstock seedlings achieved appropriate growth, they were grafted with the splice grafting technique, re-planted in 72-cell plug trays, and immediately transferred in a healing chamber.

### 2.2. Healing, Light Conditions, and Acclimatization

The healing chamber is basically a growth room where conditions are fully controlled. Specifically, the temperature was set at 25 °C, and relative humidity was initially set at 98%, gradually decreasing down to 89%, while fans ensured air circulation. The plug trays were placed on shelves irradiated with five different light wavelengths including FL (Fluora 58 W, Osram, GmbH, Munich, Germany) and LED light sources. The LEDs emitted were: (a) monochromatic red (R) with peak wavelength at 661 nm, (b) monochromatic blue (B) with peak wavelength at 450 nm, (c) an 88/12% red/blue combination (12B) which proved optimum for the healing of grafted watermelon seedlings in a recent publication of our group [7], and (d) 12B with additional 5% far-red (12B + FR) radiation with peak wavelength at 725 nm. All light treatments emitted $85 \pm 5$ $\mu$mol m$^{-2}$ s$^{-1}$ with a photoperiod of 18 h. Information about the light treatments, such as waveband percentages and the

phytochrome photostationary state, were obtained with a spectroradiometer (HD 30.1, DeltaOhm Srl, Padova, Italy) and are provided in Table 1.

**Table 1.** Wavelength distribution and photobiological parameters of the tested light treatments tested. PPS: phytochrome photostationary state. PPS was calculated according to Sager et al. [9].

| Waveband | Light Treatment | | | | |
|---|---|---|---|---|---|
| | **FL** | **R** | **B** | **12B** | **12B + FR** |
| UV %; 380–399 nm | 0 | 0 | 0 | 0 | 0 |
| Blue %; 400–499 nm | 35 | 0 | 100 | 12 | 12 |
| Green %; 500–599 nm | 24 | 0 | 0 | 0 | 0 |
| Red %; 600–699 nm | 37 | 100 | 0 | 88 | 83 |
| Far-red %; 700–780 nm | 4 | 0 | 0 | 0 | 5 |
| PPS | 0.82 | 0.89 | 0.51 | 0.89 | 0.88 |

Following the successful healing stage which lasted six days, the grafted seedlings were moved in a greenhouse for a two-week period of acclimatization where the minimum temperature was set at 21.5 °C. At this stage, the seedlings were considered commercial product and were ready for transplantation in the field.

### 2.3. Field Cultivation

Field cultivation was conducted in the experimental farm of the Laboratory of Vegetable Crops, in Aristotle University of Thessaloniki, Greece (N 40.536; E 22.995), in 2021. Following a typical analysis, the soil was characterized as sandy clay loam (SCL), moderate to heavy type. Organic matter constituted 2.3%, the pH was 7.8, and the electrical conductivity was 0.80 mS/cm. Prior to transplantation, the farm soil was plowed and crumbled, while fertilization, irrigation, and control of weeds and pathogens were in accordance with local practices. Specifically, basal dressing was conducted using a fertilizer (500 kg per hectare) including 20-5-20 (nitrogen–phosphorus–potassium) + 3 units of magnesium. During the cultivation period, plants were also fertigated twice using potassium nitrate (13.5–0–46). Irrigation was conducted depending on temperature and precipitation. Typically, the plants were irrigated every two days since precipitation was very low and the temperature was high at all times. Weeds were regularly hoed for about a month until the vines expanded and made it difficult to walk through the plants without damaging them. Finally, a few proactive crop dustings were conducted for aphids and soilborne pathogens (i.e., fusarium).

Sixteen plants from every light treatment were transplanted on 2 June 2021 over four rows considered as replicate, with a row distance of 3 m. The distance between plants within a row was 1 m. The experimental design was a randomized complete block (RCBD) with four replicates (rows). Within each row, plants were arranged in groups of four consecutive plants per light treatment and each light treatment was represented once in each row.

### 2.4. Determinations

Vegetative growth was evaluated for the first two weeks after transplanting until flowering initiation. Every week leaf number was measured, while stem diameter was determined with a digital caliber.

Flowering initiated 19 days after transplanting (DAT), but only male flowers were detected at that time. When the first female flowers bloomed, they were numbered and labeled every two days (until DAT 36) in order to identify the treatment and flowering date of the produced fruits. Specifically, female flowers were recorded on DAT 24, 26, 28, 30, 32, 34, and 36. Total, average, and gradual sum of female flowers were calculated for each measuring date.

Watermelon requires about 40 days between flowering and fruit maturity depending on the environmental conditions. At the end of the experiment, fruits were harvested

separately for each of the recorded flowering dates. For example, fruits labeled on DAT 30 were harvested after 40 days, i.e., on DAT 70. The same procedure was applied for each flowering date recorded. The few flowers that bloomed on DAT 24 did not leaf to produced fruits. Total yield and fruit number were calculated for each harvest date.

Three fruits per light treatment were evaluated regarding their biochemical content. A refractometer (PAL-α, Atago, Tokyo, Japan) was used for the determination of total soluble solids (°Brix). The Singleton and Rossi [10] method was used for the measurement of total phenolics. Antioxidant capacity was measured with the ferric reducing antioxidant power (FRAP) method according to Benzie and Strain [11]. Lycopene and total carotenoid contents were measured according to Luterotti et al. [12].

### 2.5. Statistical Analysis

Data were statistically analyzed with analysis of variance (ANOVA) using the IBM SPSS software (SPSS 23.0, IBM Corp., Armonk, NY, USA). Mean comparisons were conducted with the Scott–Knott method [13], using the statistical software StatsDirect v.2.8.0. (StatsDirect, Ltd., Birkenhead, UK) at significance level a = 0.05. The unique characteristic of this method is that it does not present overlapping in its grouping results.

### 3. Results and Discussion

Upon seedling transplantation in the field, vegetative growth was evaluated until initiation of flowering. From DAT 0, stems were significantly narrower under the influence of FL and B, an effect which carried on up to DAT 14 (Figure 1A). This is in accordance with a previous study of our group which showed that monochromatic B limited the stem diameter of acclimatized watermelon seedlings compared to red-containing LEDs [14], while FL also induced narrow stem development of the final product (unpublished observation). Stem diameter has been proposed and has widely been used as an indicator of seedling quality in vegetable species, such as tomato, pepper, eggplant [15], cucumber [16], and watermelon [17].

At the beginning of the experiment, all seedlings had an identical number of leaves (four). However, by DAT 14 12B and 12B + FR enhanced the leaf formation compared to FL, B, and R (Figure 1B). Four tomato genotypes exhibited greater leaf number under an 88/12% red/blue treatment, which is similar to our 12B, and it was concluded that the addition of blue light increases plant development and biomass production [18]. Furthermore, supplemental blue LED lighting with high-pressure sodium lamps increased fresh and dry weight and the leaf area of cucumber transplants and enhanced their development [19]. In three artichoke cultivars, blue light negatively affected the leaf number compared to red [20], while no effect was found in cucumber seedlings [21]. Tomato transplants treated with red or red–blue and red–white combinations and pepper transplants treated with high ratios of red light developed fewer leaves before the first cluster [22]. In a study with lettuce irradiated by different light sources at the seedling stage, the authors reported greater mature yield and quality when the seedlings were treated with a red/blue ratio of 2.2 compared to 1.2 or fluorescent lamps [23].

Regarding floral evaluation, male flowers started to bloom on DAT 19. On DAT 24, the first female flowers started to bloom mainly with 12B and 12B + FR and secondarily with R. With FL and B, the first female flowers bloomed on DAT 26. In the following days, flower number sharply increased up to a maximum which was on a different day for each light treatment. Specifically, with 12B and 12B + FR, flowering peaked on DAT 32, while with R, FL, and B flowering peaked on DAT 34 (Figure 2A). On every DAT, sum female flower number was significantly smaller with FL and B compared to the red-containing LEDs, with 12B + FR (mainly) and 12B (secondarily) showing the greatest values on almost every DAT (Figure 2B).

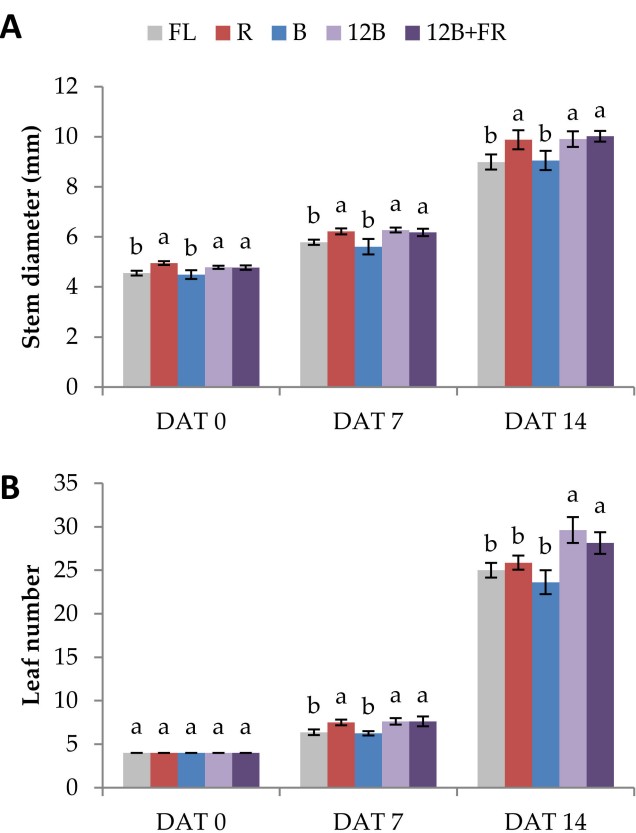

**Figure 1.** (**A**) Stem diameter and (**B**) leaf number of watermelon plants until 14 days after transplanting in the field. The seedlings were treated for six days in a healing chamber with five light treatments. Mean values ($n = 8$; ±SE) within a row followed by different letters are significantly different ($\alpha < 0.05$).

Plants possess a suite of protein photoreceptors found in the model plant Arabidopsis, which are triggered by red and far-red (phytochromes), blue (cryptochromes, phototropins, and zeitlupe group), and ultraviolet (UVR8) wavelengths [24,25]. Photoreceptors are involved in flowering through a FLOWERING LOCUS T gene, whose expression is regulated by light quality [26,27]. Far-red in particular has been found to drive the expression of the *FvFT1* gene and trigger flowering in strawberry [28]. In petunia, flowering was also promoted by far-red light under two PPFDs (98 and 288 μmol m$^{-2}$ s$^{-1}$) [3], an effect also reported in other long-day plants [29]. In a study with cucumber, red + blue light generated higher biomass, growth rate, and average internode distance in comparison with red + blue + yellow light, while the latter light treatment led to increased sucrose content, which promoted the production of female flowers [30]. In our case, 12B + FR emits red–blue light including 5% far-red, a spectrum which obviously triggered floral development earlier than FL and the monochromatic R and B wavelengths. Strikingly, only 5% far-red of 12B + FR induced flowering to a greater extent compared to 12B, pointing to the considerable effect of red/far-red ratio.

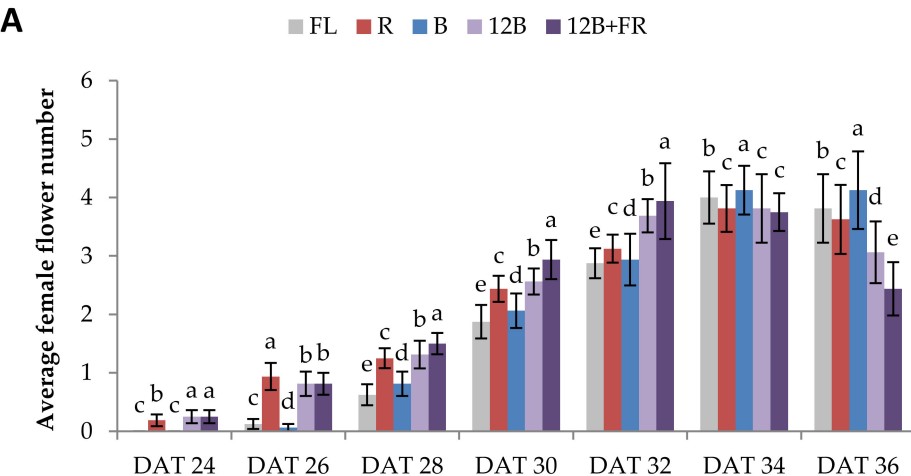

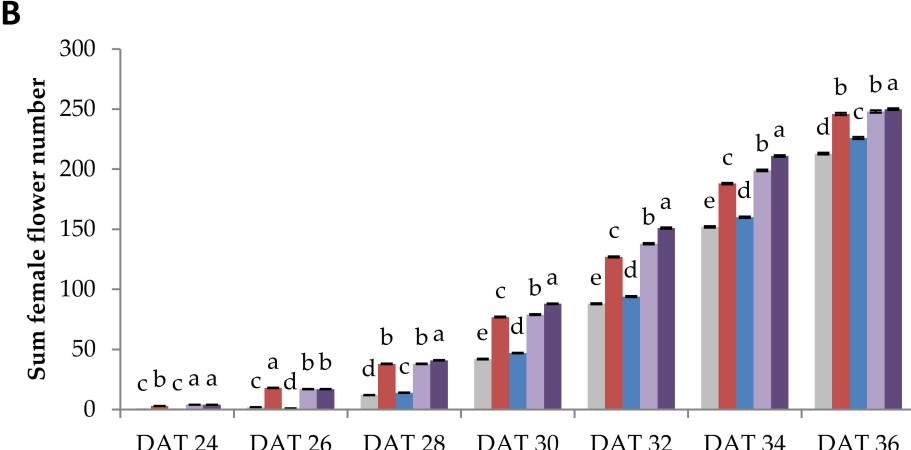

**Figure 2.** (**A**) Average female flower number on each date and (**B**) sum of female flower number of watermelon plants from the 24th to the 36th day after transplanting in the field. The seedlings were treated for six days in a healing chamber with five light treatments. Mean values ($n = 8$; ±SE) within a row followed by different letters are significantly different ($\alpha < 0.05$).

As far as fruit production is concerned, flowers that bloomed on DAT 24 did not lead to fruits in any light treatment. The first fruits were produced with R, B, 12B, and 12B + FR from flowers that bloomed on DAT 26. From DAT 28 onward, sum yield and fruit number were significantly greater with R, 12B, and 12B + FR compared to FL and B. By DAT 34, total yield and fruit number were similar for every light treatment even though there was a tendency for reduced values with B and FL (Figure 3A,B). In general, fruit production followed the pattern of flowering, indicating that fruit set was similar in plants of all light treatments.

In other cucurbits, an Italian landrace of *Cucumis melo* L. called 'Carosello leccese', grown under red + blue + far red, and red + blue LEDs, resulted in higher growth rate in comparison to plants grown under natural light spectra. Moreover, the higher number of fruits harvested and the higher water content of the fruits resulted in a 27% higher yield in total for the plants grown under LEDs [31]. Another study showed that the yield of cucumber plants grown under LEDs was higher than those grown under high-pressure sodium lamps and those grown under the combination of the two [32]. Brazaitytė et al. [8] reported that cucumber transplants treated with various light wavelengths did not exhibit after-effects on yield, but the beginning of flowering and harvest were significantly affected. Tomato and pepper transplants showed greater rates of first cluster formation and first yield when treated with blue–red combinations [22].

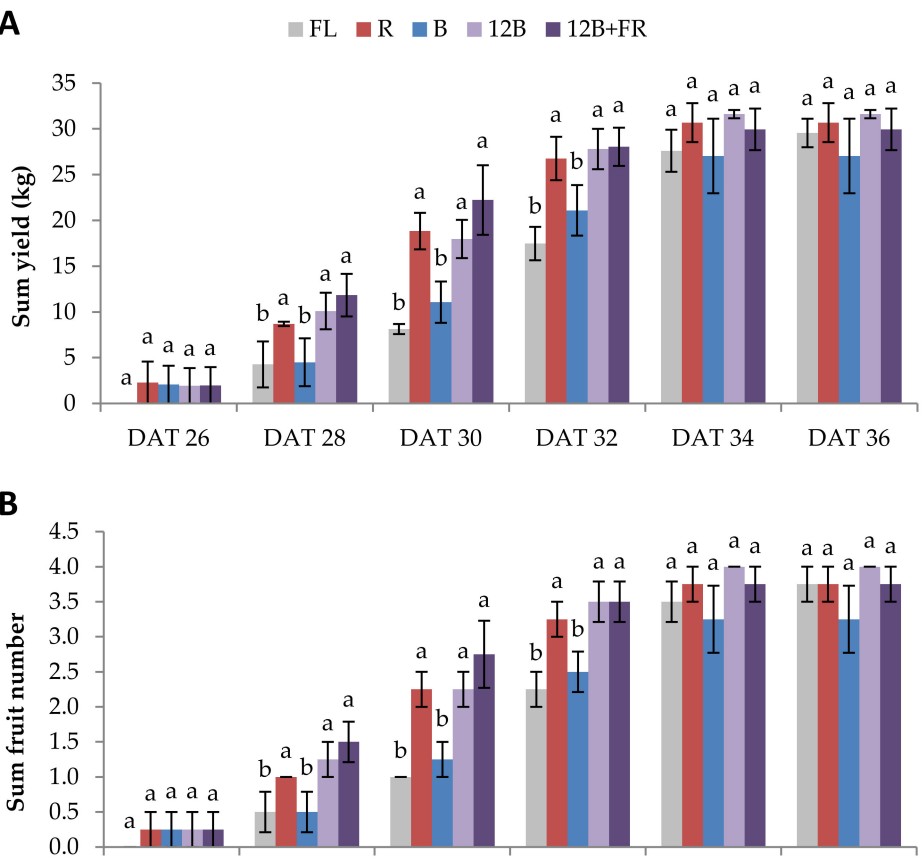

**Figure 3.** (**A**) Sum of watermelon fruit yield and (**B**) sum of fruit number derived from flowers bloomed from the 26th to the 36th day after transplanting in the field. The seedlings were treated for six days in a healing chamber with five light treatments. Mean values ($n$ = 4; ±SE) within a row followed by different letters are significantly different ($\alpha$ < 0.05).

At the end of the experiment, fruits were evaluated regarding their morphological and biochemical attributes. Specifically, the fruit length, width, and rind thickness were similar under all light treatments (Table 2). The rind/mesocarp thickness was also similar in all cases (data not shown). The same observation was made for biochemical compounds, such as total soluble solids, total phenolics, total carotenoids, lycopene, and antioxidant compounds (FRAP), which were not significantly affected by the different light treatments (Table 2).

**Table 2.** Morphological and biochemical parameters of ripe watermelon fruits after field cultivation. The seedlings were treated for six days in a healing chamber with five light treatments. Mean values ($n$ = 3; ±SE) within a row followed by different letters are significantly different (a < 0.05).

| Parameters | Light Treatments | | | | |
|---|---|---|---|---|---|
| | FL | R | B | 12B | 12B + FR |
| Length (cm) | 23.50 ± 1.26 a | 34.83 ± 2.42 a | 30.17 ± 2.17 a | 34.67 ± 0.93 a | 30.00 ± 2.02 a |
| Width (cm) | 21.67 ± 0.33 a | 20.00 ± 0.29 a | 20.50 ± 0.29 a | 19.67 ± 0.67 a | 20.67 ± 0.73 a |
| Rind thick. (cm) | 0.80 ± 0.10 a | 0.73 ± 0.13 a | 0.93 ± 0.07 a | 0.87 ± 0.09 a | 0.77 ± 0.07 a |
| TSS (°Brix) | 11.53 ± 0.13 a | 11.87 ± 0.22 a | 11.50 ± 0.35 a | 11.27 ± 0.27 a | 11.20 ± 0.42 a |
| TPC (mg/g) | 0.20 ± 0.01 a | 0.21 ± 0.01 a | 0.20 ± 0.01 a | 0.23 ± 0.01 a | 0.20 ± 0.01 a |
| TCC (µg/g) | 24.45 ± 3.54 a | 24.51 ± 0.91 a | 24.20 ± 1.10 a | 25.77 ± 2.51 a | 26.88 ± 1.14 a |
| LC (µg/g) | 19.11 ± 4.74 a | 17.89 ± 1.87 a | 18.88 ± 1.47 a | 20.60 ± 3.00 a | 21.26 ± 0.92 a |
| FRAP (µg/g) | 87.34 ± 1.39 a | 81.46 ± 1.66 a | 76.19 ± 1.94 a | 87.88 ± 3.25 a | 79.39 ± 2.35 a |

Typically, watermelon fruits can be harvested about 40 days after flowering. Summer 2021 was exceptionally stressful for outfield crops in Greece, with very high temperatures for successive weeks. Even though there were not significant differences among our treatments, fruits exhibited considerable quality in terms of sweetness (total soluble solids above 11 °Brix) and antioxidant capacity. Javanmardi and Emami [22] reported greater total soluble solids in tomato fruits treated with blue light at the seedling stage [22], but this was not evident in our case. In another study of our group [14], watermelon transplants treated with varying spectral compositions of red and blue light, and then cultivated on the field in 2018, maintained fruit quality and high quality of important nutritive characteristics. Compared to 2018, our 2021 watermelons produced about 30% greater total phenolics (0.12–0.15 mg/g in 2018 versus 0.20–0.23 mg/g in 2021) and about 50% greater antioxidant capacity displayed by FRAP (35–40 µg/g in 2018 versus 76–87 µg/g in 2021), pointing to the harsh climatic conditions which acted positively towards watermelon fruit quality. In the same study in 2018, proper morphology, root development, photosynthesis, and high fruit quality were noted on grafted watermelon plants that were treated with blue + far-red LED light during the healing stage [14].

## 4. Conclusions

The varying light qualities obviously affected the seedlings during the critical stage of tissue healing leading to certain responses. In general, 12B and 12B + FR enhanced the vegetative growth, which was evaluated until flowering initiation. Flower buds differentiated during the nursery growth when the seedlings were irradiated in the healing chamber. The same light treatments induced earlier flowering compared to the rest of the treatments. A slight addition of only 5% far-red in 12B + FR enhanced flowering compared to 12B, pointing to the significant effect of this particular wavelength for flowering. Subsequently, yield and number of fruits of 12B and 12B + FR, along with R, peaked earlier compared to B and FL. Total yield and fruit number were similar for all treatments. Overall, B had a similar response to FL throughout the experimental period with inferior vegetative growth as well as later flowering and fruit production. After 70 days from transplantation and 84 days from the seedlings' exposure to the different light qualities, no differences were detected in fruit morphological and biochemical properties. It is concluded that a light recipe including red, blue and far-red wavelengths during the healing of grafted seedlings enhances the overall growth, flowering, and yield earliness of watermelon crops. Combined with results from nursery experiments which are presented in other publications, these findings highlight the effectiveness of 12B and 12B + FR wavelengths during healing to produce high-quality grafted watermelon seedlings with greater potential for vegetative growth and rapid flower blooming and fruiting in the field.

**Author Contributions:** Conceptualization, methodology, and data analysis: F.B. and A.K.; experimental measurements: F.B., A.G. and C.D.; writing—original draft preparation: F.B. and A.G.; writing—review and editing: F.B., A.G., C.D. and A.K.; supervision and project administration: A.K. All authors have read and agreed to the published version of the manuscript.

**Funding:** This research has been cofinanced by the European Union and Greek national funds through the Operational Program Competitiveness, Entrepreneurship and Innovation, under the call RESEARCH—CREATE—INNOVATE (project code: T1EDK-00960, LEDWAR.gr).

**Conflicts of Interest:** The authors declare no conflict of interest.

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
