# Peer review of "A Light Recipe including Far-Red Wavelength during Healing of Grafted Watermelon Seedlings Enhances the Floral Development and Yield Earliness"

_agriculture, doi:10.3390/agriculture12070982_

Round 1

Reviewer 1 Report

The manuscript submitted for review was well thought out and consistently written. Very good reception of the manuscript read and the recipient's high interest in the results of the research. The aim of the research is clearly defined and the hypotheses made are understandable and supported by conclusions. The literature introduction is good and reflects the state of the current research well. The research results are presented clearly and well described. Any questions that arise during the reading are explained later in the manuscript, which is considered a positive feature of the text. I have minor comments on the research methodology. However, not for the substantive preparation of the experiment, but for its description and legibility. Overall, I rate the manuscript very well. The results obtained in the experiment are cognitively important, and may contribute to the improvement of watermelon fruit production technology and others in which the grafted plant healing procedure is used.

I have minor comments:

(Line 95-98): Were any measurements taken during the 14 day adaptation?

(Line 107-108) Fertilization, irrigation and weed control: please give a brief description.

(Line 119-128) please provide a more detailed description of the examination dates (DAT), and therefore the collection dates, as it raises some confusion when reading.

(Line 228) Did the measurements have a negative effect on the peel / mesocarp ratio of the fruit?

(Line 250-266) Conclusions: Please indicate a utilitarian conclusion from the conducted research. They are very interesting and I think they could contribute to the development of watermelon production.

Author Response

The manuscript submitted for review was well thought out and consistently written. Very good reception of the manuscript read and the recipient's high interest in the results of the research. The aim of the research is clearly defined and the hypotheses made are understandable and supported by conclusions. The literature introduction is good and reflects the state of the current research well. The research results are presented clearly and well described. Any questions that arise during the reading are explained later in the manuscript, which is considered a positive feature of the text. I have minor comments on the research methodology. However, not for the substantive preparation of the experiment, but for its description and legibility. Overall, I rate the manuscript very well. The results obtained in the experiment are cognitively important, and may contribute to the improvement of watermelon fruit production technology and others in which the grafted plant healing procedure is used.

Response: The authors would like to express their gratitude to the reviewers for the time they invested for assessing our manuscript.

I have minor comments:

(Line 95-98): Were any measurements taken during the 14 day adaptation?

Response: Indeed, a number of measurements were taken at this stage, but they are presented in another publication of our group.

Bantis, F.; Dangitsis, C.; Siomos, A. S.; Koukounaras, A. Light Spectrum Variably Affects the Acclimatization of Grafted Watermelon Seedlings While Maintaining Fruit Quality. Horticulturae 2021, 8, 10. https://doi.org/10.3390/horticulturae8010010

(Line 107-108) Fertilization, irrigation and weed control: please give a brief description.

Response, L114-122: Fertilization, irrigation and weed/pathogen control are now described in the manuscript.

(Line 119-128) please provide a more detailed description of the examination dates (DAT), and therefore the collection dates, as it raises some confusion when reading.

Response, L133-147: A more detailed description of the examination/collection dates was provided in the text.

(Line 228) Did the measurements have a negative effect on the peel / mesocarp ratio of the fruit?

Response, L254-255: The rind/mesocarp thickness was similar in all cases. It is now specified in the manuscript.

(Line 250-266) Conclusions: Please indicate a utilitarian conclusion from the conducted research. They are very interesting and I think they could contribute to the development of watermelon production.

Response, L294-298: A concluding remark highlighting the utilization potential of our findings was included in this section as suggested.

Reviewer 2 Report

This manuscript needs thorough and relevance discussion. 

Add latest and year specific area, production and productivity

Line 52- here discuss more about climate change and make it relevant to your manuscript  

Author Response

This manuscript needs thorough and relevance discussion.

Response: The authors would like to express their gratitude to the reviewers for the time they invested for assessing our manuscript. The discussion was enhanced as suggested. Please, see L179-184, L242-243, L263-265.

Add latest and year specific area, production and productivity.

Response, L33-36: This information was included in the manuscript as suggested.

Line 52- here discuss more about climate change and make it relevant to your manuscript.

Response, L57-60: The climate change implications were discussed even more, as suggested.